# Study on Local to Global Radiometric Balance for Remotely Sensed Imagery

**Xiaofan Liu [1], Guoqing Zhou [2,3,*], Wuming Zhang [4] and Shezhou Luo [5]**

1 Bo Wen College of Management, Guilin University of Technology, Guilin 541004, China; 1020180986@glut.edu.cn
2 GuangXi Key Laboratory of Spatial Information and Geomatics, Guilin University of Technology, Guilin 541004, China
3 College of Geomatics and Geoinformation, Guilin University of Technology, Guilin 541004, China
4 School of Geospatial Engineering and Science, Sun Yat-sen University, Guangzhou 510060, China; zhangwm25@mail.sysu.edu.cn
5 College of Resources and Environment, Fujian Agriculture and Forestry University, Fuzhou 350002, China; luosz@fafu.edu.cn
* Correspondence: gzhou@glut.edu.cn; Tel.: +86-773-589-6073

**Abstract:** Due to the difference of factors, such as lighting conditions, shooting environments, and time, there is compound brightness difference between adjacent images, which includes local brightness difference and radiometric difference. This paper proposed a method to eliminate the compound brightness difference of adjacent images after mosaicking, named local to global radiometric balance. It includes the brightness compensation model and brightness approach model. Firstly, the weighted average value of each row and column of image are calculated to express the brightness change; secondly, according to weighted average value, the brightness compensation model is built; thirdly, combined with the blocking method, the brightness compensation model is applied to image. Based on the value after above process, the brightness approach model is established to make the gray value of adjacent images approach to the mosaic line. In the paper, the standard deviation, MSE (mean square error) and mean value are used as the measure indices of the effect of brightness balance. The three groups of experimental results show that compared with the brightness stretch method, the histogram equalization method and the radiometric balance method, the local to global radiometric balance method not only realizes compound brightness balance, but also has better visual effects than others.

**Keywords:** compound brightness difference; brightness compensation model; brightness approach model; blocking method; brightness balance

## 1. Introduction

In order to meet different requirements, such as production of seamless data, researchers have to mosaic multiple remotely sensed images, which are collected in different orbital altitudes of satellite, from different sensors, and/or on different dates [1]. Duo to various conditions, such as solar conditions, shooting environments, sensor characteristics, etc., these images expose a number of radiometric differences from scene to scene and local brightness difference [2]. As a result, it is difficult somewhat to achieve a satisfied local to global brightness balance, seamless lines, and completely mosaicked image production [3].

For this reason, most researchers have made many efforts to minimize radiometric differences between adjacent images, which is also global brightness balance. The radiometric balance can be categorized into two types of method: (1) the blur path method; and (2) the histogram equalization method [4,5].

To achieve image brightness balance using the blur path method, researchers firstly select the overlap area in images after mosaicking; and then stretch brightness of the

area [6]. This method includes the radiometric balancing and the brightness stretching. Zhou et al. [1] proposed the radiometric balancing to eliminate radiometric difference of image. This method adjusts the mean value of adjacent images, and uses the principle of weight distribution to make the darker area obtain a higher brightness and the lighter area obtain a lower brightness, so as to achieve brightness balance. However, affected by the weight distribution, this method cannot avoid gray scale blocking [7]. The brightness stretching achieves brightness balance of the image by a fixed formula. This method includes linear brightness stretching and non-linear brightness stretching. Linear brightness stretching adjusts the gray scale of image to achieve brightness balance by changing the slope of the linear formula. The linear brightness stretching is relatively simple, but cannot solve the complicated brightness status of optical images [8]. Therefore, the non-linear brightness stretching is proposed to eliminate the image status with multiple brightness types. The non-linear brightness includes: trigonometric function, gamma function, etc. However, non-linear brightness stretching changes the brightness of image based on a fixed formula [9].

To achieve brightness balance using the histogram equalization method, the histogram of the target image needs to be adjusted according to the mosaic information between adjacent images; on the other hand, the effect relies on the histogram of the reference image [10]. Wang et al. [11] proposed an adaptive histogram equalization method for image to achieve brightness balance. Kim et al. [12] pointed out that adaptive histogram equalization method achieves brightness balance by changes to all possible gray values in the target image according to the reference image information. Wongsritong et al. [13] divided the histogram equalization method into global histogram equalization and local histogram equalization. When the reference area is small, we shall use the global histogram equalization method; and when the reference area is large, we shall use the local histogram equalization method. However, Wongsritong did not indicate how to define the size of the reference area. Yu et al. [14] proposed to use the overlap area between adjacent images as the reference area, with 60% as the boundary. When the overlap area is more than 60%, we shall use local histogram equalization, and when the overlap area is less than 60%, we shall use global histogram equalization. However, this method cannot achieve the brightness balance of the images when the overlap area between adjacent images is less than 25%. In order to improve the quality of the histogram of the reference image, this method sets restrictions according to the image brightness characteristics, such as the richness of image texture information, etc. But this method relies on processing a large number of images [15]. Kansal et al. [16] proposed to control the contrast of the target image by cutting the histogram of the reference image during the histogram equalization process. Although this method reduces the impact of contrast changes, it leads to the loss some grayscale information of the reference image. In order to overcome the limitation of the histogram, Zhou et al. [17] proposed to use the average value and standard deviation ratio of the target image as the limiting condition. This method does not need to specify a reference image, but the processing accuracy will be reduced.

In summary, it is necessary to know the overlapping area brightness information of adjacent images for the blur path method and histogram equalization method [18]. These methods need to change the gray value in the overlapping area to achieve brightness balance [19]. The histogram equalization method relies on the reference image to achieve brightness balance. However, the brightness balance effect is limited by the distribution of the reference image histogram. If the reference image is not specified, accuracy cannot be guaranteed [20]. The path blur method is not limited by the reference image, but the buffer zone needs to be selected according to the brightness difference of the image, and the effect is affected by fixed formula. When the buffer zone is small, this method will produce gray-scale blocking [21]. Above all, the image after mosaicking may have compound difference. However, the above methods can only achieve radiometric balance [22].

In order to solve the above problems, this paper proposes a local to global radiometric balance method. Compared with the blur path method and histogram equalization method,

this method solves the problems of relying on scenes, models, and overlapping area information. Simultaneously, the compound brightness difference of the image are all eliminated [23,24]. The main contributions of our study are summarized as follows.

1.  The proposed method takes into account the adaptive allocation of gray value, achieve the local brightness balance and, what is more, it is not affected by fixed formulas.
2.  The proposed method is independent of the reference image. In other words, the images can be balanced in the case of incomplete information, such as the lack of overlapping area information of adjacent images.

## 2. Local to Global Radiometric Balancing

The overall flowchart of the proposed method is shown in Figure 1. To eliminate compound brightness differences using the local to global radiometric balance method: (a) calculate the brightness characteristic value; (b) analysis brightness change of image; (c) block image if brightness of the image has a dramatic change, and establish the brightness compensation model for blocks to achieve local brightness balance; (d) establish the brightness approach model to make the gray values of adjacent optical images approach the mosaic line in order to achieve global brightness balance.

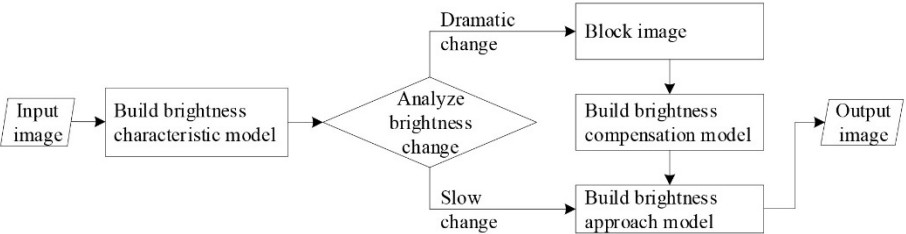

**Figure 1.** Flowchart of the proposed method.

### 2.1. Brightness Compensation Model of Local to Global Radiometric Balancing

2.1.1. Brightness Characteristic

The brightness characteristic is an important index to measure the change of image brightness. The brightness compensation model is build according to the value of the brightness characteristic to realize the local brightness balance of image.

Currently, the average gray value of an image is used to represent the brightness characteristic [25]. Setting the size of the image is $M \times N$. The brightness characteristic value of image can be expressed as follows [26]:

$$av_{(x,y)} = \frac{\sum\limits_{x=1}^{M} g_{(x,y)}}{M} \times \frac{\sum\limits_{y=1}^{N} g_{(x,y)}}{N} \tag{1}$$

where, $av_{(x,y)}$ is the brightness characteristic value of the image, $g_{(x,y)}$ is the gray value of the image.

The brightness characteristic value of the image can be calculated by using Equation (1). However, Equation (1) cannot avoid the influence of textures information of the image, resulting in errors in the brightness feature value [27]. Therefore, this paper develops the weighted average value to calculate the brightness feature value of image. The weighted average value of image brightness can be expressed as follows:

$$agv_{(x,y)} = agv_x \times agv_y = \frac{\sum\limits_{x=1}^{M} \left(g_{(x,y)} \times w_x\right)}{\sum\limits_{x=1}^{M} w_x} \times \frac{\sum\limits_{y=1}^{N} \left(g_{(x,y)} \times w_y\right)}{\sum\limits_{y=1}^{N} w_y} \tag{2}$$

where, $agv_{(x,y)}$ is the brightness characteristic value of the image, $agv_x$ is the weighted average value of image brightness in the row direction $agv_y$ is the weighted average value of image brightness in the column direction. $w_x$ and $w_y$ are the weight of the row direction and column direction of the image, respectively.

The weight of the unit pixel can be expressed as follows:

$$w_x = \frac{1}{\left| g_{(x,y)} - g_{(x-1,y)} \right|} \; w_y = \frac{1}{\left| g_{(x,y)} - g_{(x,y-1)} \right|} \tag{3}$$

where, $g_{(x-1,y)}$ and $g_{(x,y-1)}$ are the gray values of unit pixels adjacent to $(x, y)$ in the row direction and the column direction of image respectively ($1 \leq x \leq M, 1 \leq y \leq N$).

Equation (3) shows: if the gray value of adjacent pixels has slow change, the greater weight it has; If the gray value of adjacent pixels dramatic change, the lower weight it has. It should be noted, if the gray value of $g_{(x,y)}$ same to $g_{(x-1,y)}$ or $g_{(x,y-1)}$, the weight is 1.

Substituting Equation (3) into Equation (2), the brightness characteristic values are described in Equation (4):

$$agv_{(x,y)} = \frac{\sum\limits_{x=1}^{M} \frac{g_{(x,y)}}{\left| g_{(x,y)} - g_{(x-1,y)} \right|}}{\sum\limits_{x=1}^{M} w_x} \times \frac{\sum\limits_{y=1}^{N} \frac{g_{(x,y)}}{\left| g_{(x,y)} - g_{(x,y-1)} \right|}}{\sum\limits_{y=1}^{N} w_y} \tag{4}$$

### 2.1.2. The Brightness Compensation Model

Based on the brightness characteristic, the brightness compensation value can be calculated. The brightness compensation value uses the target value as a limiting condition to suppress the brightness changes in the row and column directions of the image. Yu et al. [14] proposed that the ratio of the standard deviation of the image to the coefficient of variation can be used as the target value. Therefore, the target value of the image can be expressed as follows:

$$mv = \frac{\sigma_I}{\text{cov}_I} \tag{5}$$

where, $\sigma_I$ is the standard deviation of the image, $\text{cov}_I$ is the coefficient of variation. $\text{cov}_I$ can be expressed as follows:

$$\text{cov}_I = \frac{\sum\limits_{x=1,y=1}^{M,N} \sqrt{\frac{(g_{(x,y)} - av_{(x,y)})^2}{M \times N}}}{agv_{(x,y)}} \tag{6}$$

where, $av_{(x,y)}$ is the gray average value of image which can be calculated by Equation (1).

Combining Equations (4) and (5), the brightness compensation value of the image can be expressed as follows:

$$bc_{(x,y)} = \frac{mv^2}{agv_{(x,y)}} \tag{7}$$

where, $bc_{(x,y)}$ is the brightness compensation value of image, $mv$ is the target value of image, $agv_{(x,y)}$ is the brightness characteristic value of the image.

Combining Equation (4) and Equation (7), the brightness compensation value of the image in the row direction can be expressed as follows:

$$bc_x = \frac{\left( \sum\limits_{x=1}^{M} w_x \right) \times mv}{\sum\limits_{x=1}^{M} \frac{g_{(x,y)}}{\left| g_{(x,y)} - g_{(x-1,y)} \right|}} \; y \in [1, N] \tag{8}$$

where, $bc_x$ is the brightness compensation value of image in the row direction.

Similarly, the brightness compensation value of image in the column direction can be expressed as follows:

$$bc_y = \frac{\left(\sum\limits_{y=1}^{N} w_y\right) \times mv}{\sum\limits_{y=1}^{N} \frac{g_{(x,y)}}{\left|g_{(x,y)} - g_{(x,y-1)}\right|}} \quad x \in [1, M] \tag{9}$$

where, $bc_y$ is the brightness compensation value of image in the column direction.

Above all, the principle of brightness compensation is as shown in Figure 2.

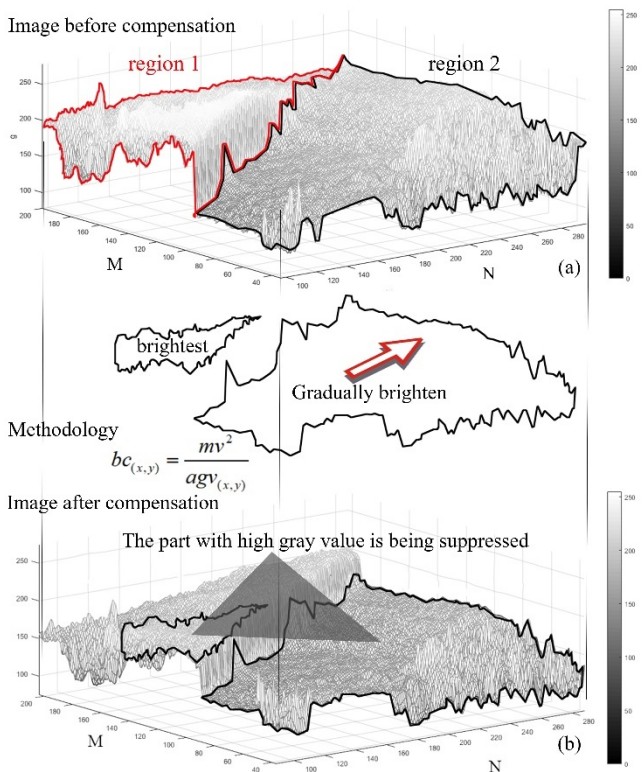

**Figure 2.** The principle of brightness compensation. (**a**) Image before compensation. (**b**) Image after compensation.

### 2.1.3. Blocking and Compensating on Image

According to Equation (9), we need (a) to obtain the weight through the difference between gray value of adjacent pixels; (b) to calculate the brightness compensation value [27]. However, Yu et al. [14] proposed that the texture of the ground features have a greater impact on the coefficient of variation. Furthermore, Chen et al. [28] proposed that the gray value difference between adjacent pixels of the image is larger, the coefficient of variation is larger, and the accuracy is lower. According to Equation (8), the coefficient of variation will affect the calculation of the brightness compensation value. Therefore, this paper uses the block and extract method to find the block(s) with lower coefficient of variation to calculate the brightness compensation value. The block and extract method should have the following principles [14].

**Principle 1.** *There is no overlap between blocks after the image is divided.*

**Principle 2.** *The rows and columns direction of the extracted block cannot appear discontinuous when the number of extracted blocks is more than one block.*

**Principle 3.** *If the number of extracted blocks is equal to the number of divided blocks, it is meaningless to use the blocking method.*

As mentioned above, using the block and extract method to divide the image into four blocks, which is divided into $2 \times 2$ parts. As shown in Figure 3, the situation will be same when the number of divided blocks is less than four. There will be a violation of principle 2 when the number of divided blocks is more than four.

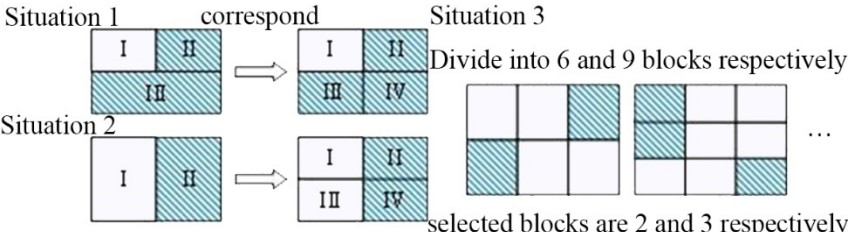

**Figure 3.** Special cases in image block.

As shown in Figure 3: in situation 1, divided image into three blocks is the same to divided image into four blocks if extract are two and three respectively. In situation 2, divided image into two blocks is same to divided image into four blocks if extract blocks are one and two respectively. In situation 3, there will be appear discontinuous when divided image into more than four blocks [29].

Let the name of blocks are $bl_1$, $bl_2$, $bl_3$, $bl_4$. In order to select the blocks with lower coefficient of variation for calculating the brightness compensation value. We extracts block(s) by setting up the threshold. The threshold can be expressed as follows:

$$ts = \frac{\sum\limits_{u=1,2,3,4} \text{cov}_u}{4} \tag{10}$$

where, $\text{cov}_u$ is coefficient of variation of $bl_u$ ($u \in [1,4]$).

According to Equation (10), corollaries are shown as follow:

**Corollary 1.** *Compared with other blocks, blocks with a high coefficient of variation will be excluded ($t_u > ts$).*

**Corollary 2.** The block with low coefficient of variation can still be selected when the coefficients of variation of each block are similar ($t_u < ts$).

According to the aforementioned, the brightness compensation value of the image is calculated by interpolation calculation.

If the blocks are extracted are $bl_2$ and $bl_3$, the extraction process as shown in Figure 4.

As shown in Figure 4. (a) the image is divided into 4 blocks; (b) according to the *ts*, the blocks are extracted; (c) the weighted average value of image brightness are calculated; Finally, according to Equation (7), the brightness compensation value are calculated.

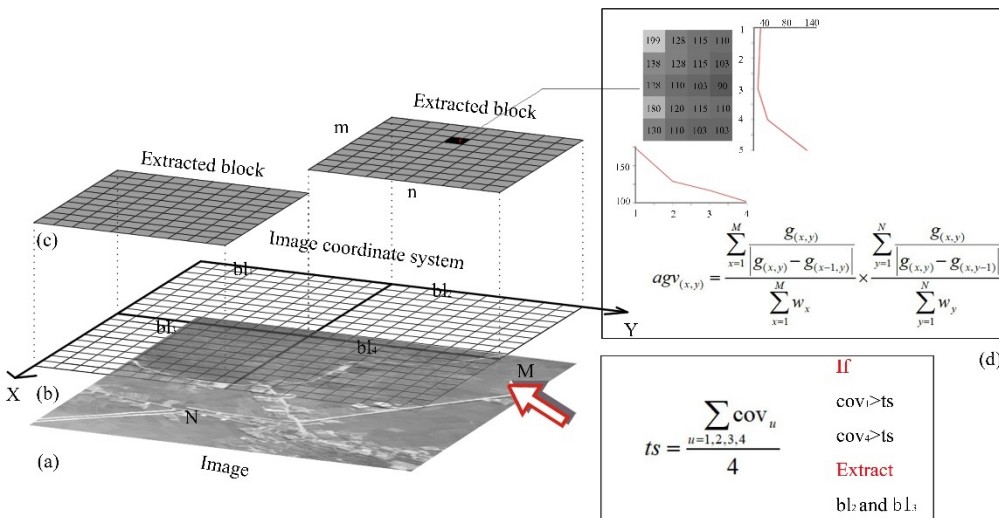

**Figure 4.** Block and extract: (**a**) original image, (**b**) creating coordinate system, (**c**) extracting blocks, (**d**) the example of calculate brightness compensation.

Above of all, setting $BC_b$ is the set of brightness compensation value of the column in the image. Setting $BC_a$ is the set of brightness compensation value of the row in the image. Then the brightness compensation model can be expressed as follows:

$$BC_I = \left[ BC_a^{bl2} \cup BC_a^{bl3} \right]^T \left[ BC_b^{bl3} \cup BC_b^{bl2} \right] \tag{11}$$

where, $BC_I$ is the brightness compensation model, and,

$$
\begin{aligned}
BC_a^{bl_2} &= \left[ \begin{array}{ccccc} bc_{a_1}^{bl_2} & bc_{a_2}^{bl_2} & bc_{a_3}^{bl_2} & \dots & bc_{a_m}^{bl_2} \end{array} \right] \\
BC_b^{bl_2} &= \left[ \begin{array}{ccccc} bc_{b_1}^{bl_2} & bc_{b_2}^{bl_2} & bc_{b_3}^{bl_2} & \dots & bc_{b_n}^{bl_2} \end{array} \right] \\
BC_a^{bl_3} &= \left[ \begin{array}{ccccc} bc_{a_1}^{bl_3} & bc_{a_2}^{bl_3} & bc_{a_3}^{bl_3} & \dots & bc_{a_{M-m}}^{bl_3} \end{array} \right] \\
BC_b^{bl_3} &= \left[ \begin{array}{ccccc} bc_{b_1}^{bl_3} & bc_{b_2}^{bl_3} & bc_{b_3}^{bl_3} & \dots & bc_{b_{N-n}}^{bl_3} \end{array} \right]
\end{aligned}
\tag{12}
$$

According to Equation (11), the image after local brightness balance process can be expressed as follows:

$$LB = BC_I \odot region \tag{13}$$

where, $LB$ is the image after local brightness balance process, "$\odot$" represents the Manhattan accumulation.

### 2.2. Brightness Approach Model of Local to Global Radiometric Balancing

The brightness compensation model is used to eliminate the local brightness differences in images with compound brightness differences. However, the radiometric difference caused by mosaic of adjacent images (the adjacent images is named image 1, 2, n) still exist. The position of the mosaic line in the image can be expressed as follows:

$$
ML = \left[ \begin{array}{cccc}
ml_{(1,2)} & & & \\
& ml_{(2,3)} & & \\
& & \ddots & \\
& & & ml_{(M,N-1)}
\end{array} \right] \tag{14}
$$

According to Equations (13) and (14), the image which has compound brightness difference after brightness compensating can be expressed as follows:

$$AI = \begin{bmatrix} g_{(1,1)} \times BC_1^L & g_{(1,2)} \times BC_1^R & \cdots & & g_{(1,C)} \times BC_{r_2}^R \\ g_{(2,1)} \times BC_1^L & g_{(2,2)} \times BC_2^L & \cdots & g_{(2,C-1)} \times BC_{r_2-1}^R & g_{(2,C)} \times BC_{r_2}^R \\ \vdots & & \vdots & & \vdots \\ g_{(Z,1)} \times BC_1^L & & \cdots & g_{(Z,C-1)} \times BC_{r1}^L & g_{(Z,C)} \times BC_{r_2}^R \end{bmatrix} \tag{15}$$

where, $AI$ is the image after local brightness balance, $BC_{r_1}^L$ is the brightness compensation value of image to the left of the mosaic line, $BC_{r_2}^R$ is the brightness compensation value of the image to the right of the mosaic line, $C$ and $Z$ are the size of the image after mosaicking.

The brightness approach value is calculated to eliminate the radiometric difference of image basis on the local brightness balance. The brightness approach value is achieving radiometric balance by making gray values the same at the mosaic line.

Firstly, the average of the mean gray value of images are regard as approach value to achieve brightness approach at mosaic line. According to Equation (1), the mean gray value after local brightness balance are described in Equation (16):

$$\mu_{(x,y)}^L = \frac{\sum\limits_{x=1,y=1}^{Z,C} g_{(x,y)} \times BC_{r_1}^L}{r_1}$$
$$\mu_{(x,y)}^R = \frac{\sum\limits_{x=1,y=1}^{Z,C} g_{(x,y)} \times BC_{r_2}^R}{r_2} \tag{16}$$

where, $\mu_{(x,y)}^L$ and $\mu_{(x,y)}^R$ are the mean gray value of images at mosaic line to the left and right respectively. $BC_{r_1}^L$ and $BC_{r_2}^R$ are the brightness compensation values, respectively.

According to Equation (16), the brightness approach value can described in Equation (17):

$$AV = \frac{\mu_{(x,y)}^L + \mu_{(x,y)}^R}{2} \tag{17}$$

where, $AV$ is the brightness approach value. According to the brightness approach value, the approach coefficients on both sides of the mosaic line can be expressed as follows:

$$t_1 = \frac{AV}{g_{(x_1,y_1)}}$$
$$t_2 = \frac{AV}{g_{(x_2,y_2)}} \tag{18}$$

where, $t_1$ and $t_2$ are the approach coefficients on both sides of the mosaic line, respectively, $g_{(x_1,y_1)}$ and $g_{(x_2,y_2)}$ are the gray value on both sides of the mosaic line, respectively.

Then, the model of brightness approach value on rows is described in Equation (19):

$$p^L = \frac{1-t_1}{1-r_1} \times Z + \frac{t_1-r_1}{1-r_1} \qquad (Z \in [1, r_1])$$
$$k^R = \frac{1-t_2}{1-r_2} \times Z + \frac{t_2-r_2}{1-r_2} \qquad (Z \in (r_1, r_1 + r_2]) \tag{19}$$

where, $p^L$ is the brightness approach value on rows, which can darken the image, $k^R$ is the brightness approach value on rows, which can brighten the image. Similarly, the brightness approach value on the column is described as $p^D$ and $k^U$.

Secondly, combined with Equations (18) and (19), the brightness approach model can be expressed as follows:

$$P = (P^D)^T P^L \qquad K = (K^R)^T K^U \tag{20}$$

where,

$$
\begin{aligned}
P^L &= \begin{bmatrix} p_1^L & p_2^L & \cdots & p_{r_1}^L \end{bmatrix} \\
P^D &= \begin{bmatrix} p_1^D & p_2^D & \cdots & p_{r_3}^D \end{bmatrix} \\
K^R &= \begin{bmatrix} k_{r_1+1}^R & k_{r_1+2}^R & \cdots & p_{r_1+r_2}^R \end{bmatrix} \\
K^U &= \begin{bmatrix} k_{r_3+1}^U & k_{r_3+2}^U & \cdots & p_{r_3+r_4}^U \end{bmatrix}
\end{aligned}
\tag{21}
$$

Setting the elements of $p^D$ and $k^U$ be 1, when the image only has the radiometric difference on column direction. Otherwise, setting the elements of $p^L$ and $k^R$ be 1.

Thirdly, make the image elements that needs to be darkened 1, and all other elements 0, and this can be expressed as follows:

$$
ILB_b = \begin{bmatrix}
1 & 0 & 0 & \cdots & 0 & 0 \\
1 & 1 & 0 & & 0 & 0 \\
\vdots & & \ddots & \ddots & & \vdots \\
1 & 1 & 1 & \cdots & 0 & 0
\end{bmatrix}
\tag{22}
$$

Combining Equations (20) and (21), the result that darkens the image is expressed as follows:

$$
GB_d = LB_b \odot P = \begin{bmatrix}
p_{m_1,n_1} & 0 & 0 & \cdots & 0 & 0 \\
p_{m_2,n_1} & p_{m_2,n_2} & 0 & & 0 & 0 \\
\vdots & & & \ddots & \ddots & \vdots \\
p_{m_M,n_1} & p_{m_M,n_2} & p_{m_M,n_3} & \cdots & 0 & 0
\end{bmatrix}
\tag{23}
$$

where, $GB_d$ is the result that darkens the image.

Similarly, the result that brightens the image can be expressed as follows:

$$
GB_b = \begin{bmatrix}
0 & k_{m_1,n_2} & k_{m_1,n_3} & \cdots & k_{m_1,n_{N-1}} & k_{m_1,n_N} \\
0 & 0 & k_{m_2,n_3} & & k_{m_2,n_{N-1}} & k_{m_1,n_N} \\
\vdots & & \ddots & \ddots & & \vdots \\
0 & 0 & 0 & \cdots & k_{m_M,n_{N-1}} & k_{m_M,n_N}
\end{bmatrix}
\tag{24}
$$

where $GB_b$ is the result that brighten the image.

Finally, the result by the brightness approach model can be expressed as follows:

$$
GB = GB_b + GB_d
\tag{25}
$$

where, $GB$ is the result by brightness approach model. Based on Figure 2b, the principle of radiometric balance can be expressed as in Figure 5.

Combining the brightness compensation model and the brightness approach model, the local to global radiometric balancing for image can be expressed as follows:

$$
GLB = GB \odot LB
\tag{26}
$$

where, $GLB$ is the result after local to global radiometric balancing.

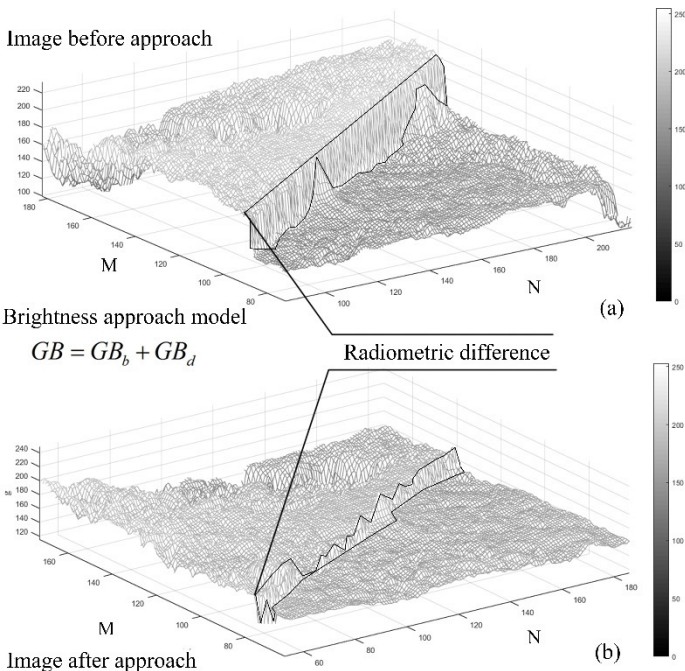

**Figure 5.** The principle of brightness approach. (**a**) Image before approach. (**b**) Image after approach.

## 3. Experiments and Results

Using local to global radiometric balancing to achieve a brightness balance of image with compound brightness difference takes place through the following 3 steps:

**Step 1.** *Block the image and select the blocks according to the threshold. The threshold can be calculated by Equation (10).*

**Step 2.** *Establishing the brightness compensation model. The model can achieve local brightness balance of image, and can be calculated by Equation (12).*

**Step 3.** *Establishing the brightness approach model. The model can achieve global radiometric balance of image, and can be calculated by Equation (20).*

### 3.1. Data Setargon Is a Kind of Satellite

As shown in Figure 6, in order to prove that the local to global radiometric balancing is suitable for brightness equalization of multiple types of image sets, we have selected three sets of images taken by the argon satellite and Landsat satellite for experiments.

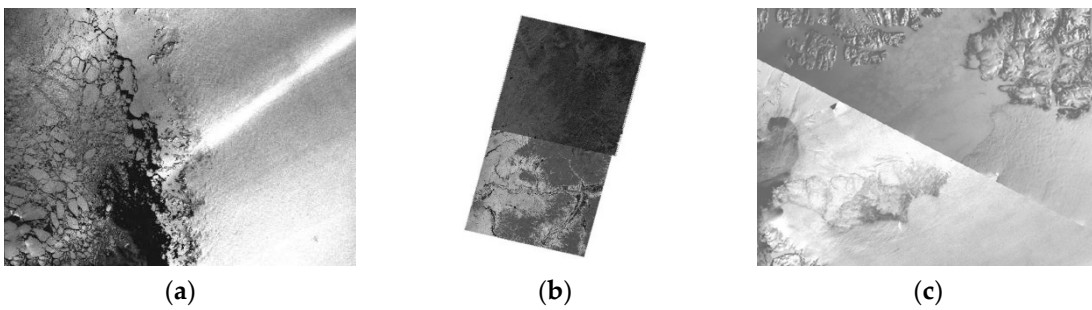

**Figure 6.** The image set for prove local to global radiometric balancing. (**a**) and (**c**) The image taken by argon KH-5. (**b**) The image taken by Landsat.

As shown in Figure 6, firstly, these images has the following characteristics:

1. The image were taken by satellites with different flying altitudes.
2. The images were collected over a long time span (1963–2011).
3. The images had different resolution.

Secondly, these images had different requirements:

1. Figure 6a has local brightness differences. The brightness compensation method needs to be used to process the image to achieve brightness balance.
2. Figure 6b has a radiometric difference. The brightness approach method needs to be used to process the image to achieve brightness balance.
3. The Figure 6c has a compound brightness difference. The local to global radiometric balancing needs to be used to achieve brightness balance.

### 3.2. The Experiments and Results

3.2.1. Local Brightness Balance

As shown in Figure 6a, the image was taken in 1967 in Greenland by ARGON KH-5. The satellite has a ground resolution of 140 m and a scan width of 556 km. The brightness compensation model is used to achieve local brightness balance of image.

The local brightness balance can be achieved by following steps:

(a) As shown in Figure 7a. Blocking the image and count the coefficient of variation according to Equation (6). The coefficients of variation of each block are shown in Table 1.

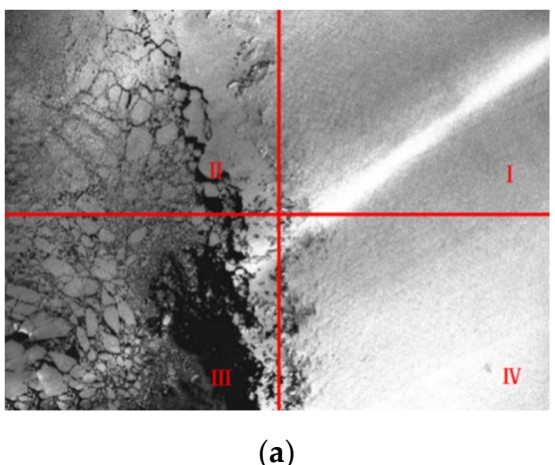 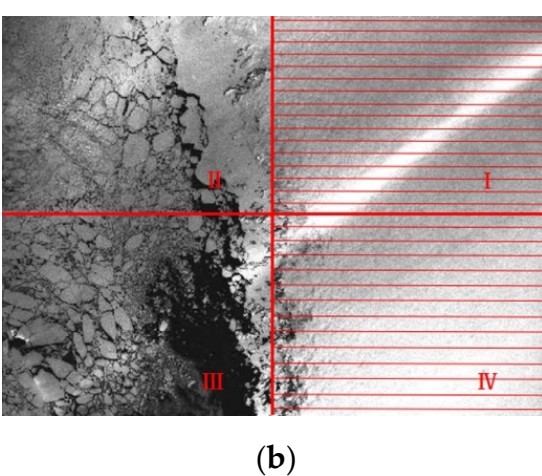

|(a)|(b)|

**Figure 7.** Blocking and selecting. (**a**) Blocking image. (**b**) Selecting blocks

**Table 1.** The coefficients of variation of each block.

|  | Block I | Block II | Block III | Block IV |
|---|---|---|---|---|
| coefficient of variation | 0.1727 | 0.1935 | 0.3119 | 0.2340 |

As shown in Figure 7b, according to the Equation (10) and combining this with the Table 1, the threshold of the image is 0.3880. Then, the blocks I and IV (I and IV as shown in Figure 7b) are selected.

(b) The weight is calculated by Equation (3) and the brightness characteristic value is calculated by Equation (4) according to the block I and IV. As shown in Figure 8, the model of brightness characteristic is established through fitting the value.

As shown in Figure 8, $avx$ is the brightness characteristic value of row direction of block I and IV, $avy$ is the brightness characteristic value of column direction of block I and IV, $n$ is the number of blocks, $M$ is the number of image. As shown in Figure 9, according to the mean value of brightness which is 0.5946, the brightness compensation model is established by Equation (7).

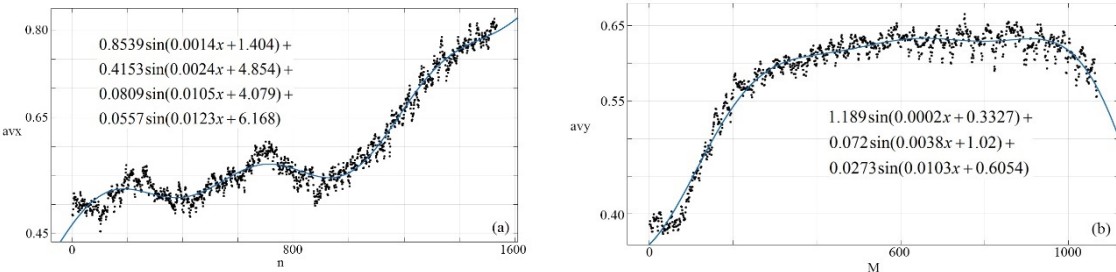

**Figure 8.** The model of brightness characteristic. (**a**) The model of row direction. (**b**) The model of column direction.

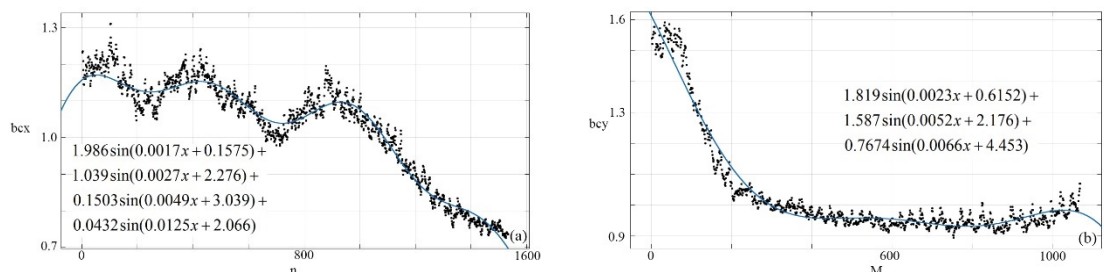

**Figure 9.** The model of brightness compensation. (**a**) The model of row direction. (**b**) The model of column direction.

As shown in Figure 9, $bc_x$ is the brightness characteristic value of row direction of block I and IV, $bc_y$ is the brightness characteristic value of column direction of block I and IV.

Finally, the image is processed combining the interpolation method and Equation (12). The result of local brightness balancing is shown in Figure 10.

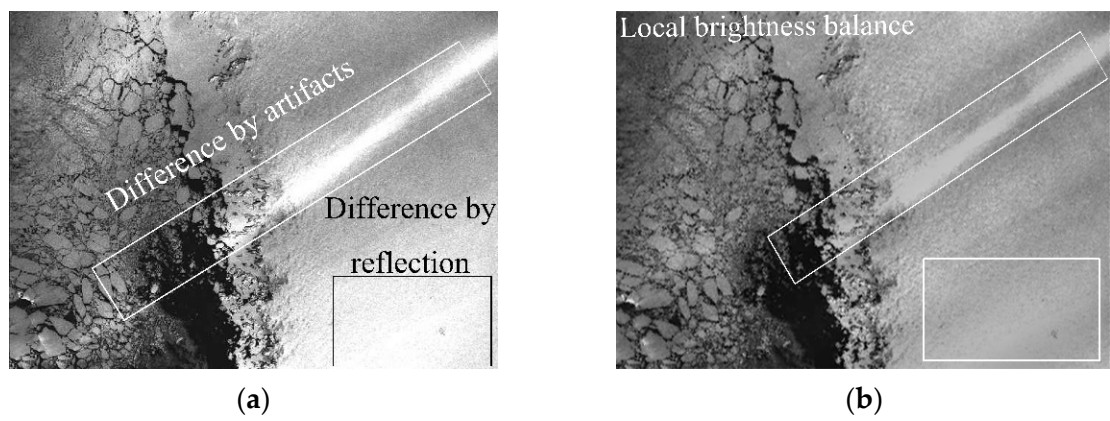

**Figure 10.** The result of local brightness balancing. (**a**) The image before process. (**b**) The image after process.

### 3.2.2. Global Radiometric Balance

The image was taken in 2003 and 2011 at Heilongjiang province, China by Landsat 3 and 4. The Landsat 3 and 4 have different sensors, different number of bands, and shooting at different times as shown in Figure 11.

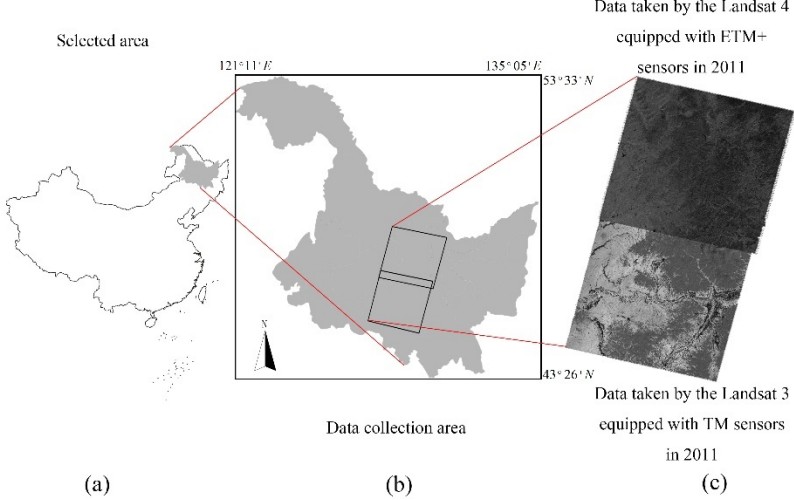

**Figure 11.** Image sources. (**a**) Selected area. (**b**) Image collection area. (**c**) Image obtained.

As shown in Figure 11, the image has obvious radiometric differences. Using the brightness approach model of local to global radiometric balancing can achieve the brightness equation and follows these steps:

**Step 1.** *Calculate the $P^D$ which can darken the image, according to Equation (19).*

**Step 2.** *Calculate the $K^U$ which can brighten the image, according to Equation (19).*

**Step 3.** *Building the model of brightness approach model, according to Equation (20).*

**Step 4.** *Process the image according to Equations (21) to (23).*

The original image and result by using brightness approach model are shown in Figure 12.

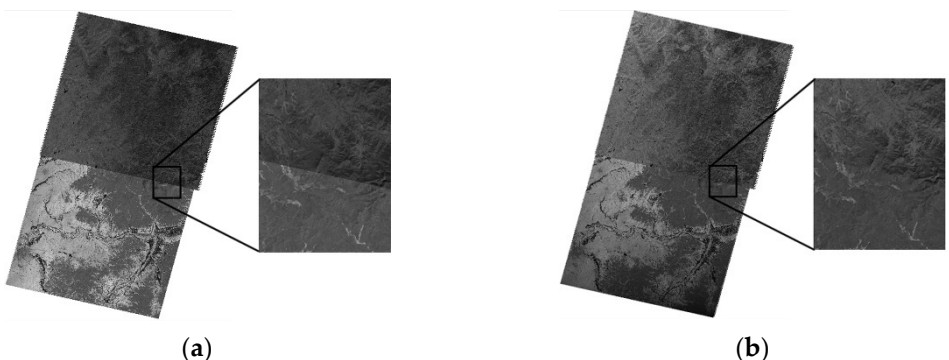

**Figure 12.** Compare with original image and the result by using brightness approach model. (**a**) Original image. (**b**) The result by using brightness approach model.

As shown in Figure 12a, the original image has obviously radiometric difference. As shown in Figure 12b. The result by using brightness approach model eliminate radiometric difference.

### 3.2.3. Compound Brightness Balance

As shown in Figure 13a, the image has compound brightness difference. Combined the brightness compensation model and brightness approach model of local to global radiometric balancing to achieve brightness balance. As shown in Figure 13b and Table 2, we block the images into 4 blocks respectively, and count the coefficient of variation of each blocks. For statistical convenience, we named the brighter region as image 1, and the darker region as image 2. It should be noted that blocks with too small area will not be counted.

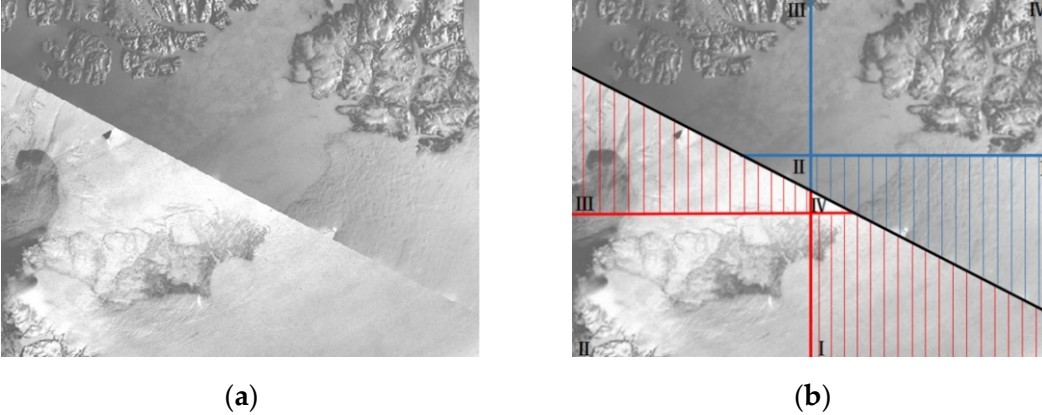

**Figure 13.** Selecting the blocks of images. (**a**) Original image. (**b**) Selected blocks.

**Table 2.** The coefficients of variation of each block.

|  | Block I | Block II | Block III | Block IV |
|---|---|---|---|---|
| Coefficient of image 1 | 0.1551 | 0.2262 | 0.1856 | Null |
| Coefficient of image 2 | 0.0357 | Null | 0.1607 | 0.2042 |

According to the Equation (10), the threshold of image 1 and 2 are 0.1890 and 0.1335, respectively. As shown in Figure 13b. The block I of image 1 and the block I and III of image 2 are selected.

As shown in Figure 14, the model of brightness characteristic and compensation of image 1 are established through fitting the value.

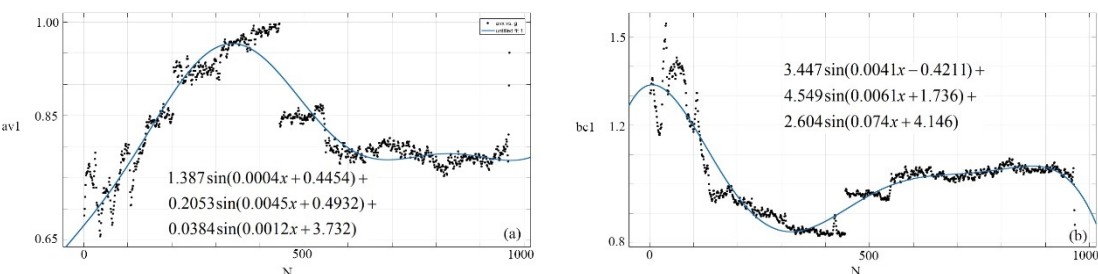

**Figure 14.** The model of brightness characteristic and compensation of image 1. (**a**) Model of brightness characteristic. (**b**) Model of brightness compensation.

where, $av_1$ is brightness characteristic value of image 1, $bc_1$ is the brightness compensation value of image 1.

As shown in Figure 15, similar to the processed of image 1, the model of brightness characteristic and compensation of image 2 are established through fitting the value.

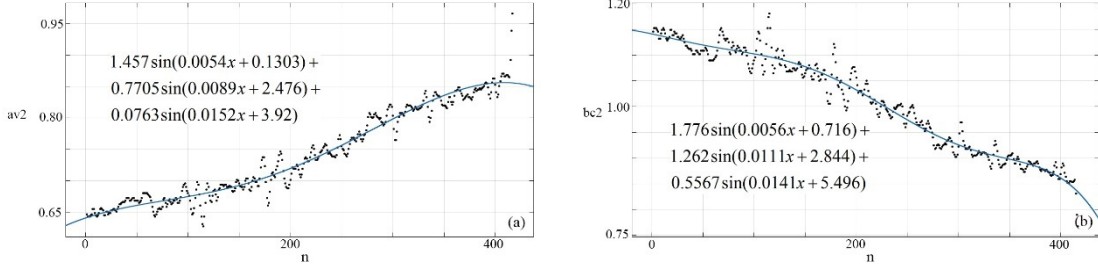

**Figure 15.** The model of brightness characteristic and compensation of region 1. (**a**) Model of brightness characteristic. (**b**) Model of brightness compensation.

where $av_2$ is brightness characteristic value of image 2, $bc_2$ is the brightness compensation value of image 2.

The whole process is shown in Figure 16. According to Equation (12), the result after brightness compensation model of local to global radiometric balancing is shown in Figure 16b. Compared with the original image, the result achieves local brightness balance.

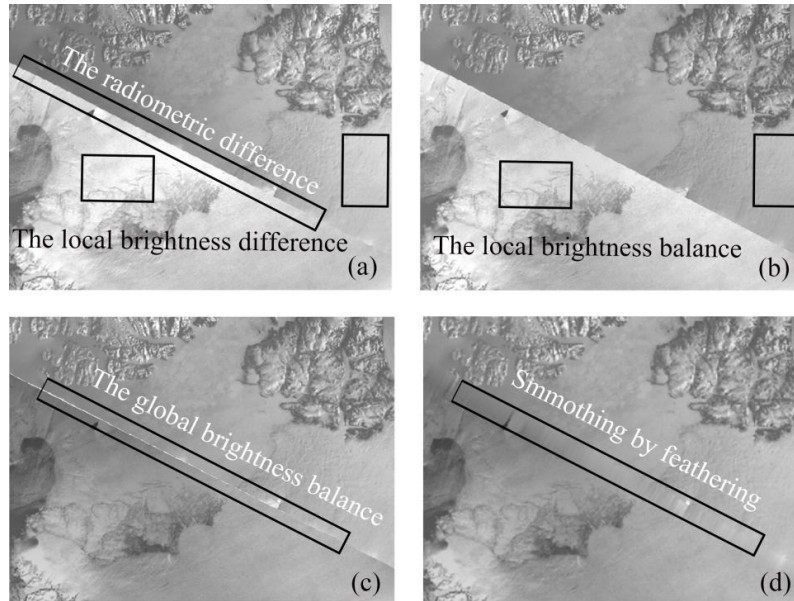

**Figure 16.** The process of local to global brightness balancing. (**a**) Original image. (**b**) The result after using brightness compensation model of local to global radiometric balancing. (**c**) The result after using brightness approach model of local to global radiometric balancing. (**d**) The result after feathering.

However, the result after the brightness compensation model of local to global radiometric balancing still has radiometric difference. Therefore, the brightness approach model of local to global radiometric balancing is used to achieve global brightness balance.

According to Equations (19) to (26), the brightness approach model is established to achieve global brightness balance. The result by using the brightness approach model is shown in Figure 16c. As shown in Figure 16d, feathering is used to smooth the result.

## 4. Discussion

Compared with the results by different methods and the original image, we discuss the effects on terms of vision and index.

### 4.1. In Terms of Vision

In the experiment of local brightness balance, as shown in Figure 17a, the original image has local brightness differences caused by artifacts and reflections. Brightness stretching is the most commonly used to achieve local brightness balance method. As shown in Figure 17b, the result by brightness stretch reduces the difference between artifacts and surrounding gray values. However, it cannot eliminate the brightness difference by reflections. As shown in Figure 17c, the result by brightness compensation method of local to global radiometric balancing has a better effect in eliminating the difference caused by reflection and image details are preserved.

In the experiment of global brightness balance, as shown in Figure 18a, the original image has obviously radiometric difference cause by taken in different times, equipped different sensors. The result by using the histogram equalization method in the case of unknown overlapping area is shown in Figure 18b. The result has a poor effect. On the one hand, it enhances the contrast of the original image. On the other hands, radiometric

difference is not eliminated. The result by using the brightness approach model of local to global radiometric balancing is shown in Figure 18c. It should be emphasized that these two experiments were conducted under the conditions of lack of overlapping area information and lack of reference images.

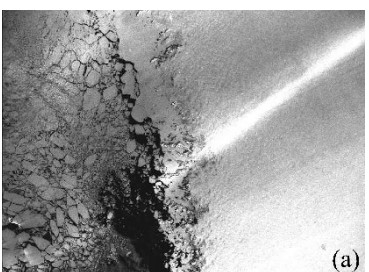 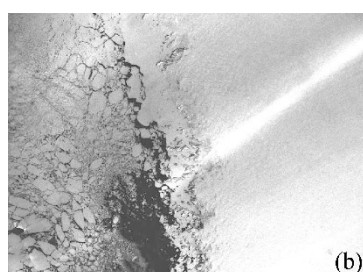 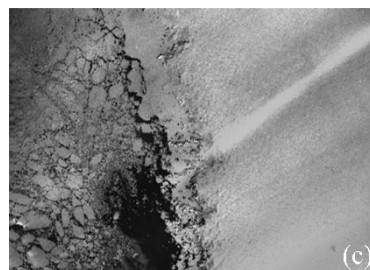

(a) (b) (c)

**Figure 17.** Comparing the result by different methods. (**a**) Original image. (**b**) Result by brightness stretch [1]. (**c**) Result by brightness compensation method.

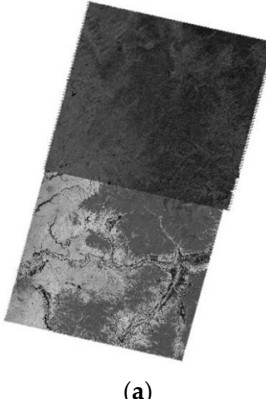 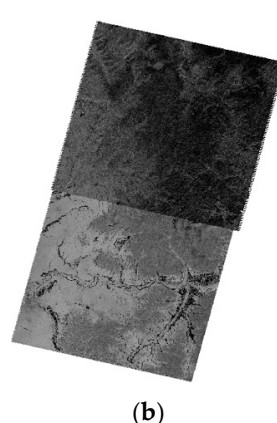 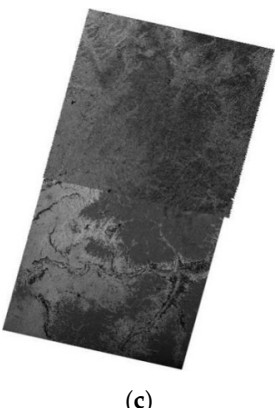

(**a**) (**b**) (**c**)

**Figure 18.** Compare with results by different method. (**a**) Original image. (**b**) Result by using histogram equalization method. (**c**) Result by using local to global radiometric balancing.

In the experiment of global brightness balance, the original image is shown in Figure 19a. The image has obvious compound brightness difference. The result by using radiometric balancing is shown in Figure 19b. The result is roughly global brightness balance. However, it has the phenomenon of gray scale blocking, which is a product due to the local brightness difference. As shown in Figure 19c, the result by using local to global radiometric balancing has better visual effect than former.

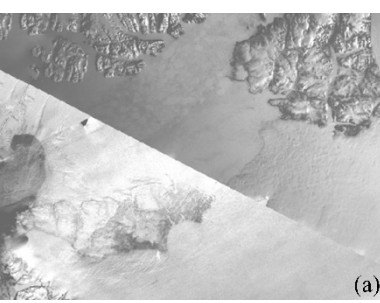 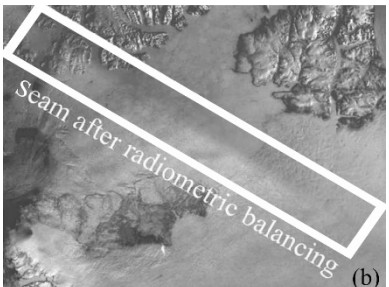 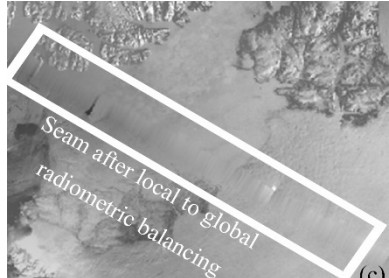

(a) (b) (c)

**Figure 19.** Compare with results by different method. (**a**) Original image. (**b**) Result by using radiometric balancing [1]. (**c**) Result by using local to global radiometric balancing.

### 4.2. In Terms of Index

The index includes standard deviation, MSE and mean value. As shown in Table 3. The standard deviation is an index to express the contrast. The MSE is an index to express the degree of deviation from the original image. The mean value is an index to express the situation of gray value compare with original image. Compare with the index of results by different methods, we can known:

**Table 3.** The index of results by different methods.

| Experiments | | Standard Deviation | MSE | Mean Value |
|---|---|---|---|---|
| Experiment of local brightness balance | Original image | 0.2465 | Null | 0.59202 |
| | Result by brightness stretch | 0.1771 | 0.0464 | 0.7567 |
| | Result by local to global radiometric balancing | 0.1790 | 0.0333 | 0.5096 |
| Experiment of global brightness balance | Original image | 0.1750 | Null | 0.5827 |
| | Result by histogram Equalization method | 0.1825 | 0.0205 | 0.5596 |
| | Result by local to global radiometric balancing | 0.1207 | 0.0225 | 0.5554 |
| Experiment of compound brightness balance | Original image | 0.1986 | Null | 0.7118 |
| | Result by radiometric balancing | 0.1693 | 0.0439 | 0.5942 |
| | Result by local to global radiometric balancing | 0.1042 | 0.0348 | 0.6470 |

In the experiment of local brightness balance the standard deviation of the result by the brightness compensation model of local to global radiometric balancing are reduce 0.0675 than original image (0.2465), and is similar to the standard deviation of the result by brightness stretching (the difference is 0.0019). The result by the brightness compensation model has lower MSE than the result by brightness stretching (the difference is 0.0131) and the mean value of the result by the brightness compensation model is similar to original image.

In the experiment of global brightness balance the standard deviation of the original image is 0.1750. The standard deviation of the result by using the histogram equalization method is 0.1825, in the case of unknown overlapping area. The standard deviation of the result by using the brightness approach model of local to global radiometric balancing is 0.1207, in the case of the unknown overlapping area. These indexes show that the result by the brightness approach model of local to global radiometric balancing has ba better effect when the mean and standard deviation value are similar.

In the experiment of compound brightness balance the standard deviation of the original image is 0.1986. The standard deviation of result by using radiometric balancing is 0.1693. The standard deviation of the result by using local to global radiometric balancing is 0.1042. Compared with the original image, the standard deviation of the result by using local to global radiometric balancing is reduced by 0.0944. However, the standard deviation by using the radiometric balancing is reduced by 0.0293. In addition, the result by local to global radiometric balancing has lower MSE than the result by radiometric balancing, and mean value is closer to the original image. Therefore, the local to global radiometric balancing is better than radiometric balancing.

### 5. Conclusions

We have proposed the local to global radiometric balancing to eliminate compound brightness difference. The brightness compensation model of local to global radiometric balancing is used to achieve local brightness balance. Compared with brightness stretching, the result by local to global radiometric balancing has lower MSE, and mean value is closer to the original image. The brightness approach model of local to global radiometric balancing is used to achieve radiometric balance. Compared with the histogram equalization

method, the result by local to global radiometric balancing has lower standard deviation. The local to global radiometric balancing has lower standard deviation and lower MSE than radiometric balancing when processing the image with compound brightness difference. The main innovation of our study is the proposed method, which is not limited by the information of overlapping area and reference images.

**Author Contributions:** Conceptualization, G.Z. and X.L.; methodology, X.L.; software, X.L.; validation, W.Z. and S.L.; resources, G.Z.; writing—original draft preparation, X.L.; supervision, G.Z., W.Z. and S.L.; project administration, G.Z.; funding acquisition, G.Z. All authors have read and agreed to the published version of the manuscript.

**Funding:** This research was funded by the Guangxi universities scientific research basic capacity improvement project numbers 2021KY1670; National Natural Science of China under Grant numbers 41431179, 41961065; Guangxi Innovative Development Grand Grant under the grant number: GuikeAA18118038, GuikeAA18242048; the National Key Research and Development Program of China under Grant numbers 2016YFB0502501 and the BaGuiScholars program of Guangxi (Guoqing Zhou).

**Acknowledgments:** The author would like to thank the reviewers for their constructive comments and suggestions.

**Conflicts of Interest:** The authors declare no conflict of interest.

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
