# Peer review of "Study on Local to Global Radiometric Balance for Remotely Sensed Imagery"

_remotesensing, doi:10.3390/rs13112068_

Round 1

Reviewer 2 Report

This paper proposes a strategy to eliminate the compound brightness difference after mosaicking adjacent images. This proposed methodology includes the brightness compensation model and the brightness approach model. The authors compare with other methodologies, a set of experimental data.

I found the paper to be interesting and the contribution seems clear enough in the Introduction part. The mathematical basis is correct. The findings are interesting and the topic is relevant to the journal. The paper is well structured and mostly well written but the needs some linguistic revisions and some editing revisions.

Some specific concerns are as follows:

1) In Figure 3, the figure 3d and in general, the equations are not easy to see (the example of calculate brightness compensation).  

2) In line 309, it should be indicate if the first image corresponds to a), and the second one to b)…..

3) In line 487 I think it is very risky to say as a conclusion “the proposed method can deal with muti-types data such as the image taken in different sensors, different resolution”. I think it is not sufficiently justified.

Round 2

Reviewer 1 Report

All my concerns have been answered.

Author Response

Thank you very much for your professional advice. Your opinion really makes the overall structure, content and theoretical basis of my article more detailed. Thanks again for your guidance.

Reviewer 2 Report

Accept in present form

Author Response

Thank you very much for your professional advice. Your opinion really makes the overall structure, content and theoretical basis of my article more detailed. Thanks again for your guidance.

This manuscript is a resubmission of an earlier submission. The following is a list of the peer review reports and author responses from that submission.